# Changes in Quality of Life in Treatment-Resistant Schizophrenia Patients Undergoing Avatar Therapy: A Content Analysis

**DOI:** 10.3390/jpm13030522

**Published:** 2023-03-14

**Authors:** Mélissa Beaudoin, Stephane Potvin, Kingsada Phraxayavong, Alexandre Dumais

**Affiliations:** 1Department of Psychiatry and Addictology, University of Montreal, Montreal, QC H3T 1J4, Canada; 2Faculty of Medicine and Health Sciences, McGill University, Montreal, QC H3G 2M1, Canada; 3Research Center of the University Institute in Mental Health of Montreal, Montreal, QC H1N 3V2, Canada; 4Services et Recherches Psychiatriques AD, Montreal, QC H1N 3V2, Canada; 5Institut National de Psychiatrie Légale Philippe-Pinel, Montreal, QC H1C 1H1, Canada

**Keywords:** schizophrenia, Avatar Therapy, quality of life, processes of change, auditory verbal hallucinations, content analysis

## Abstract

Avatar Therapy has a significant impact on symptoms, beliefs, and quality of life of patients with treatment-resistant schizophrenia. However, little is known about how these changes are implemented into their lives and to which aspects of their lives these improvements relate. Ten consecutive patients enrolled in an ongoing clinical trial were assessed using semi-guided interviews before as well as three months after Avatar Therapy. These encounters have been recorded and transcribed so that the discourse could be thoroughly analyzed, leading to the generation of an extensive theme grid. As the cases were analyzed, the grid was adapted in a back-and-forth manner until data saturation occurred. The content analysis allowed the identification of nine main themes representing different aspects of the patients’ lives, each of which was subdivided into more specific codes. By analyzing the evolution of their frequency, it was observed that, following therapy, patients presented with fewer psychotic symptoms, better self-esteem, more hobbies and projects, and an overall improved lifestyle and mood. Finally, investigating the impact of Avatar Therapy on quality of life allows for a deeper understanding of how people with treatment-resistant schizophrenia can achieve meaningful changes and move towards a certain recovery process.

## 1. Introduction

Auditory verbal hallucinations are amongst the most prevalent and invalidating symptoms of schizophrenia [1]. Hearing voices, especially persecutory ones, can have a significant impact on one’s quality of life (QoL), notably by limiting their activities (e.g., unemployment) and affecting their interpersonal relationships [1,2]. The amount of time invested in care can also be considerable, not to mention the direct and indirect costs at the individual and societal level [3,4]. To this day, antipsychotic medication is and remains the mainstay of treatment for psychotic symptoms, including auditory verbal hallucinations [5,6]. However, even when treated with appropriate doses, a significant proportion of patients fail to see significant improvement [5,7,8,9]. In these cases, it is recommended that patients undergo psychotherapy as an adjunct to their pharmacological treatment [8,10]. Although cognitive-behavioral therapy for psychosis (CBTp) is currently the most studied and widespread psychotherapy, it is only moderately effective on symptoms and has little to no effect on patients’ functioning and QoL [10,11]. Other therapeutic options include brain stimulation treatments, including electroconvulsive therapy, which are usually used as a last resort [12]. Although they can be effective for many patients, these methods are invasive, not always readily available, and some of their possible benefits might not be sustained over time [13,14].

QoL is defined by the World Health Organization as “an individual’s perception of their position in life in the context of the culture and value systems in which they live and in relation to their goals, expectations, standards and concerns” [15]. In health science, an increasing amount of importance is given to this concept as it might better represent the recovery processes than solely relying on symptoms [16,17,18]. Indeed, improvements in psychiatric symptomatology do not always translate to actual change in an individual’s life, or in their perception of how they experience life [19]. Over time, multiple tools have been developed to quantitatively measure QoL [20,21,22]. Common indicators include wealth, employment, education, activities, social life, security, freedom, as well as mental and physical health [20,21,22].

Avatar Therapy for psychosis (ATp) is a novel psychotherapy using virtual reality (VR) to treat auditory verbal hallucinations in treatment-resistant schizophrenia [23,24,25]. This approach allows the patients to enter a dialogue with their most distressing voice, which is represented by an Avatar in a virtual environment. The Avatar starts by being confrontational to the patient, but the focus of therapy quickly shifts to reconciliation, self-esteem, and consolidation. Although ATp is still at an experimental stage, it was shown to effectively reduce hallucinations (distress and frequency) and beliefs about voices [23,24]. A randomized, single-blinded, and controlled clinical trial is currently in progress to demonstrate the superiority of ATp over CBTp (ClinicalTrials.gov (accessed on 28 January 2023) Identifier: NCT04054778).

Unlike CBTp, ATp has also been shown to induce a significant increase in participants’ QoL [23,24]. Indeed, a recent clinical trial conducted by our research team has shown a 10% increase in QoL (measured using the Quality of Life Enjoyment and Satisfaction Questionnaire—Short Form; [26]) when comparing the three-month post-therapy evaluation to baseline (*p* < 0.001) [24]. However, standardized tools have their limitations, as these do not capture all the richness of such a complex concept. Commonly used scales such as the EQ-5D or the SF-6D have been criticized for not appropriately assessing the QoL of people with mental health disorders [27,28]. For this reason, qualitative research remains relevant to identify how people with these complex illnesses see the concept of quality of life, and what precisely impacts it from their perspective [29]. So far, none of the validated scale covers all the aspects of QoL that were found to be important to people with severe mental illnesses in the literature [28]. Regarding ATp specifically, in light of previously published results on QoL, it remains indeed unclear which aspects of therapy impact the participants’ lives (e.g., activities, relationships, etc.) and, most importantly, whether these changes are subjectively meaningful for them. To better understand how psychotherapeutic interventions work, it is important to ask patients directly about their experience. A qualitative research design is particularly relevant to answer these exploratory questions and guide future research.

Although the therapeutic processes of ATp have been explored [30], it remains uncertain how ATp translates into real changes in the patients’ lives. Nevertheless, several hypotheses can be formulated. Notably, it has been suggested that psychotherapeutic interventions should target both rational restructuring and emotional arousal [31,32]. It is indeed plausible that ATp does both, as this therapy is not only experiential and emotion-inducing, but the beliefs and self-perceptions of the patient are also questioned by the Avatar; therefore, cognitive processes are also an important focus of therapy [30]. Since everything occurs in a safe space, under the control and supervision of a therapist, these techniques allow patients to incorporate new information and change their relationships with their voices, thereby reducing anxiety and increasing self-confidence [30,33]. Another factor accounting for ATp efficacy could be the use of Avatars. A study investigating the effects of animations on learning processes suggested that the use of an Avatar allows the development of a relationship that is more appealing, persuasive and longer lasting [34]. Subsequently, a more positive relationship with the voices could lead to a better mood, a better motivation to achieve changes, and eventually an improvement in QoL.

To better understand how these changes occur, and more specifically what exactly is changing, it is essential to explore the patients’ perspectives. To do so, an in-depth qualitative analysis of their discourse might allow a better understanding of how what is learned during therapy translates into real life. Therefore, this study aims to qualitatively explore changes in the QoL of patients who underwent ATp. Thus, it is hoped that our understanding of the implementation of clinically meaningful changes in schizophrenia will be improved.

## 2. Materials and Methods

ATp is a novel psychotherapy using virtual reality to treat auditory verbal hallucinations in treatment-resistant schizophrenia [23,24,25]. This approach allows voice hearers to create an Avatar representing their most distressing voice. First, patients are invited to personalize the Avatar’s physical appearance as well as its voice (e.g., pitch). Then, they are encouraged to enter a dialogue with their Avatar, which is animated by the therapist, while being immersed in a virtual environment by wearing a head-mounted VR headset. During each subsequent therapeutic session, the patient and the therapist first discuss what happened during the past week and work together to define the goals of the current session. Then, around 15 min is allocated to immersion in VR, during which the patients are invited to converse with their Avatar. At the end of each session, the therapist discusses with the patient what has just happened, and goals are set for the coming week. During the first VR sessions, the Avatar mainly repeats what the patient reports hearing on a daily basis, which is generally mainly negative and denigrating. During these sessions, the therapist encourages the patient to reply in an assertive manner. Eventually, over the course of the sessions, the Avatar gradually starts opening up to the patient, questioning their beliefs, and agreeing with the patient. Additional details about each specific therapeutic session as well as the interactions between the patients and their Avatar have been described elsewhere [30].

The first ten consecutive patients who were assigned to receive ATp in the context of an ongoing randomized clinical trial (ClinicalTrials.gov (accessed on 28 January 2023) Identifier: NCT03585127) and who had completed a three-month follow-up interview were enrolled in the present study. To be included in the trial, participants needed to be at least 18 years old and having been diagnosed with either schizophrenia or schizoaffective disorder, which was confirmed using the Structured Clinical Interview for DSM-5 (SCID-5; [35]). Moreover, all of them underwent at least two trials (minimum six weeks) of different antipsychotic medications without satisfactory responses. Additionally, their medication had to have been stable for at least two months prior to their enrollment, their physical and mental health both had to be stable, and their medication regimen was not allowed to change during the trial. Of all the participants included in this clinical trial, none of them declined to participate in this particular qualitative subproject. Once the initial evaluation was completed, each patient underwent nine weekly sessions of approximately one hour, during which about 15 min were allocated to the immersion in VR with their Avatar. Two semi-guided qualitative interviews were conducted by one of two interviewers: the first one occurred two weeks prior to the first therapy session, and the second one was scheduled three months after the last therapy session. Two interviewers, both experienced research nurses, underwent rigorous training and conducted multiple interviews in the presence of the principal investigator (AD) before the start of this study. For comparability purposes, each patient was interviewed by the same person before and after therapy. During these interviews, the participants were first invited to talk about their life in general. The interview then continued in the format of a discussion, mainly following the patient’s lead. Nevertheless, the interviewers made sure the following subjects were covered in a very exhaustive manner, prompting the patient with open-ended questions when necessary: interpersonal relationships, interests, work, activities, and personal goals. Moreover, the interviewer also made sure each of the following aspects, included in the Maastricht Interview, a tool to assess voice hearing, were covered: nature of the experience, characteristics of the voices, personal history of hearing voices, triggers, content, beliefs about the origin of the voices, impacts on the patient’s life, balance of the relationship, coping strategies, experience of childhood, medical history, and social networks [36]. This structured questionnaire, designed to assess the voice hearing experience, included open-ended questions that could be used when the information did not come out spontaneously in the patient’s speech. Indeed, some patients needed to be prompted with specific questions more often than others, especially when negative symptoms were more prominent. Finally, to avoid inducing biases, questions regarding Avatar Therapy specifically were kept for the very end of the post-therapy interview. Patients were notably asked if and how therapy impacted their lives. As these in-depth interviews could sometimes trigger unpleasant memories or feelings in some participants, measures were put in place to support the participants as much as possible. Notably, participants were given the space to express their feelings and take breaks. Moreover, they all had access to 24/7 phone assistance through which they could talk to a qualified nurse, who could redirect them to the appropriate resources if necessary.

A total of 20 interviews (10 before and 10 after the therapy) were transcribed, and the transcription was counter-validated by a different person. A detailed content analysis was then conducted based on widely used methods [37,38,39,40]. First, each transcript was carefully read and annotated by two different members of the research team, which consisted of seven annotators (including MB), and then thoroughly discussed in a group format (4 to 10 people) to identify emerging themes. Links between these themes were identified to group them into broader meaningful categories. This process allowed for the creation of a preliminary theme grid, as well as a dictionary including all the themes with their definitions and associated examples. Among the authors, MB, KP, and SP participated in this analysis—along with 12 student contributors.

Before starting the coding process, each transcript was first segmented by ideas (a verbatim was defined as a single idea) by MB. Then, each verbatim was assigned to a code (one of the identified themes) by two independent members of the research team, one interview at a time. Each disagreement was carefully revised by the two coders and, when a consensus could not be reached, these were discussed in a group format with all the investigators and contributors. It was then decided if and how the theme grid should be modified, either by adding a new code, changing the definition of an existing one, reorganizing the grid or adjusting the idea segmentation. At the end of this process, an agreement was reached to classify each verbatim into a single code. This process was repeated until data saturation was reached, which was defined by a stable theme grid for at least three consecutive transcripts.

When all the transcripts were coded, a group discussion including all investigators and an experienced psychiatrist (AD) took place to decide which themes were of interest regarding QoL and should be included in the present article. For example, neutral and descriptive utterances (e.g., I have a sister) were excluded, whereas those that were determined to be positive (e.g., my sister is very supportive) or negative (e.g., my sister is mean to me) were retained. Moreover, many of these codes were merged based on clinically relevant similarities.

Coding of the transcripts allowed the generation of the frequency of appearance of each theme, pre-therapy vs. post-therapy, as well as the number of patients for whom these themes were present. Only recent and present verbatims relating to present or recent ideas were included in the analyses, except for the “Wishes” theme, for which the verbatims referring to the future were included as well. Moreover, all verbatims that were directly guided or greatly influenced by the interviewer (e.g., the patients repeated something the interviewer said or answered by yes or no) were also excluded. The evolution of each patient’s QoL was first qualitatively described, and then quantitative counting allowed the confirmation of these impressions.

The frequency of each code as well as the number of participants for whom each code was present before and after therapy were extracted using QDA Miner version 5.0.25, a qualitative analysis software designed by researchers to organize, annotate, retrieve, and analyze collections of documents and images [41]. Additionally, QDA Miner generated an agreement between the two annotators for each interview: the Scott’s Pi. This statistic usually varies between 0 and 1, and a higher value represents a better agreement. Negative scores are also possible, suggesting no agreement or that the agreement is lower than chance. The following benchmarks may be used to assess the strength of agreement: <0.00 = Poor; 0.00–0.20 = Slight; 0.21–0.40 = Fair; 0.41–0.60 = Moderate; 0.61–0.80 = Substantial; 0.81–1.00 = Almost perfect [42]. Finally, each verbatim and its associated code was exported into Excel to allow the calculation of a variation score for each code. The purpose of this score was to consider the intra-patient variations in theme frequency. For each subtheme, each patient was attributed a score of 1 if the frequency was higher during the post-therapy interview compared to pre-therapy, a score of 0 if it remained the same, or −1 if the frequency decreased. Then, these 10 scores were added together to obtain an overall score for each sub-theme ranging from −10 to 10. Consequently, a negative score meant that this subtheme was overall less discussed post-therapy compared to before, whereas a positive score meant that the subtheme was brought up more often by patients following therapy.

## 3. Results

### 3.1. Sample Characteristics

Baseline characteristics of the sample were presented in Table 1. Most of the participants were unemployed, single Caucasian males with diagnoses of schizophrenia, mostly treated with clozapine (making them “ultra-resistant” to pharmacological treatment) [43].

Participants were initially assessed through baseline interviews, which were conducted between December 2019 and November 2021, and which lasted between 56 and 93 min (average: 71 ± 16 min). Case #3’s interview occurred right before the start of the pandemic, and therefore there was a significant delay (130 days) before the start of therapy. However, excluding this case, interviews occurred on average 17 days before the start of therapy.

### 3.2. Qualitative Analysis of Change

Self-reported changes which occurred during and following ATp were reported in Table 2. Information presented in this table was all collected during the semi-directed interviews occurring pre-therapy (general information about the patient, baseline assessment, patient expectation regarding therapy) as well as post-therapy (assessment of changes, what their lives are like now, their satisfaction, and finally how did they think therapy had an impact), which were conducted between October 2020 and February 2021, and which lasted between 34 and 61 min (average: 48 ± 7 min). These interviews were conducted on average 101 days (3.3 months) following the end of therapy, which corresponded to 222 days (7.3 months) after the first interview. However, the COVID-19 pandemic began between the two interviews for the first three participants. Due to containment measures, these had a much higher average elapsed period between the two interviews (314 ± 16 days) compared to the other seven participants, for whom the baseline interview occurred during the pandemic (182 ± 17 days).

By performing this qualitative analysis, it was possible to observe that eight participants out of ten experienced important changes to their lives. Notably, seven of them had a drastic diminution of their voices (cases #1, 2, 3, 5, 8, 9, 10), and the remaining participant (case #4) mentioned that they no longer succeeded in making them angry. All of them mentioned feeling better, and three of them saw a reduction in their anxiety (cases #1, 2, 3). Four participants had new projects (cases #1, 3, 4, 5), and two even managed to achieve an objective that they had established prior to therapy (cases #8 and 9). However, no major changes were noted for cases #6 and #7.

### 3.3. Content Analysis

A total of 20 interviews were analyzed, which allowed the identification of nine main themes: psychiatric symptomatology, occupations, interpersonal relationships, identity, wishes, lifestyle, psychiatric care, life events, and attitudes/behaviors during the interview. All these themes included many more subthemes, which allowed the categorization of the entirety of the interviews. For the current study, only the subthemes that were considered relevant in regard to the participant’s QoL were presented in Table 3. For example, details about the participants’ pasts (e.g., a story that happened many years ago) were excluded from the analysis.

The first theme included everything related to the patients’ psychiatric symptomatology, as well as their perceptions and beliefs regarding their symptoms. It was subdivided into three main subthemes: psychotic symptoms, impact of symptoms on QoL, and beliefs. Overall, three months after therapy, patients displayed fewer psychotic symptoms as opposed to prior to therapy. Moreover, the remaining symptoms were perceived more positively. Many participants also directly stated that their symptoms had less impact on various aspects of their QoL following ATp.

Secondly, the Occupation theme included everything related to how the patients occupied their time: their professional lives, studies, hobbies, routine, personal care, etc. It was divided into two categories: work/studies and daily activities. In summary, participants seemed to have more occupational activities following therapy. However, these changes mainly concerned hobbies and chores, and not so much work/study-related activities.

The third theme included all verbatims about interpersonal relationships. It included improvement strategies, the nature of these relationships, and perceptions of change from those close to them. To sum up, a few changes were noted regarding the interpersonal relationships of a minority of participants. Nevertheless, it does not seem to be a major impacted component following ATp.

The fourth theme, Identity, included all verbatims related to how patients perceived themselves: their interests, personality traits, skills, and how they related to themselves. It was divided into four subthemes: interests, personality, skills, and relation to oneself. In summary, patients displayed more self-appraisal during the post-therapy interview. They were also more prone to talk positively about their skills, and three of them even experienced reconciliation with themselves.

The fifth theme included all the wishes patients had regarding their futures, projects, goals, steps taken, etc. Overall, patients had more personal projects following therapy compared to before. Additionally, they were more likely to have taken steps toward achieving them.

The sixth theme comprised any verbatim related to the patients’ lifestyles. This included anything related to substance consumption and addictions, physical health, mood, and housing. In summary, patients presented with an overall improved lifestyle following therapy. Notably, they had fewer addictive behaviors, better moods, and sometimes even better housing situations. Regarding physical health, some participants mentioned having healthier habits (e.g., better sleep) and they talked less about their disorders and symptoms.

The seventh included all that was related to the patients’ psychiatric care, whether it was related to ATp or to the regular follow-up by their treatment team. Overall, ATp might have impacted how some participants perceived their regular treatment team. Moreover, all of them mentioned that ATp directly helped them in some way.

Second to last, the Life Events theme included all the utterances describing major life events as well as everything related to the COVID-19 pandemic. Neutral verbatims were excluded. No participant-specific life event had a major impact on their QoL. However, the COVID-19 pandemic impacted all of them. Nevertheless, good adaptations and positive aspects of the pandemic were noted during the post-therapy interviews only. Finally, during the semi-directed interview, the patients often expressed themselves about how they were currently feeling or mentioned that the voices were present during the interview.

Delusional, disorganized, or fixed thoughts often hindered data collection, which led to the creation of the Attitudes/Behavior during the interview theme. This theme brings together the different attitudes and behaviors that have been observed: presence of hallucinations, patient cooperation, and confusion. Overall, attitudes and behaviors did not significantly hinder the interviews, except for case #6. Indeed, his delusional and disorganized speech greatly impaired data collection during the post-therapy interview.

### 3.4. Quantitative Summary of Changes

Each verbatim was classified in a specific subtheme (code) using the QDA Miner software. This process was done by two independent annotators, and according to Scott’s Pi, the agreement was generally moderate (Scott’s Pi of 0.4–0.6 for 17/20 interviews) and, rarely, slight (Scott’s Pi of 0.2–0.4 for 3/20 interviews). Finally, each disagreement was thoroughly discussed in a group format until an agreement was reached.

The frequency of apparition of each code as well as the number of participants for whom this code was present, before and after therapy, was calculated using QDA Miner; these statistics were presented in Table 4. The variation score corresponds to the number of participants for whom the frequency increased between the baseline and the three-month interview, minus the number of participants for whom it decreased.

## 4. Discussion

ATp is a novel psychotherapy which has recently been shown to reduce auditory verbal hallucinations as well as improving QoL in patients with treatment-resistant schizophrenia. Using VR, patients were invited to engage in a dialogue with their most distressing voice, which is represented by an Avatar animated by the therapist. In doing so, it is believed that the relationship between the patients and their voice(s) is improved, which would translate into meaningful changes to their daily life. To determine how this happens, this study aimed to explore the patients’ subjective perception of how ATp impacted their QoL. By performing an extensive content analysis, we were able to identify the main themes qualifying the patients’ QoL before as well as three months after therapy, thus providing a deeper understanding of how ATp impacts their lives. Overall, following therapy, it was observed that patients presented with fewer psychotic symptoms, and that the remaining ones were perceived more positively. Moreover, they seemed to have better self-esteem, more projects, new hobbies, and an overall improved lifestyle and mood.

First, nine major themes were identified: psychiatric symptomatology, identity, occupations, wishes, interpersonal relationships, lifestyle, psychiatric follow-up, life events, and attitudes/behaviors during the interview. Each theme was subdivided into much more specific codes that characterized the ideas that were questioned or brought up by the patients themselves. For example, a theme that was not systematically questioned but that often came up anyway was “Consumption/Addictions”. Behaviors and attitudes during the interview were also coded as it was determined that these could hinder the course of the interview, and therefore impact code frequency. Nevertheless, most of these themes were consistent with the different topics that had to be covered by the interviewer, and therefore they correspond to our initial expectations.

The primary aim of ATp is to improve auditory hallucinations; therefore, it is not surprising that important changes were observed in the psychiatric symptomatology. Notably, the intensity of these symptoms diminished, the patients’ feelings regarding them got more positive, the relationship between the patients and their respective voices improved and delusions were less present. Indeed, dialoguing with voices, which is encouraged in ATp, could be less distressing than having command or malevolent voices, as well as persecutory delusions [44]. The patients were also less prone to believe that their voices were omnipotent following therapy. This could also be a mechanism of ATp, as literature shows that developing control over auditory verbal hallucinations could increase patients’ functioning and reduce distress [45]. Moreover, many patients directly stated the symptoms had less impact on their QoL.

The patients’ perception of their identity also seemed to evolve following therapy, notably by displaying signs of better self-esteem. Indeed, following therapy, patients were much less self-deprecating. Instead, they talked more positively about themselves by mentioning their strengths or skills. Improving self-perceptions is one of the main goals of ATp [30], and therefore, this change was expected. This therapy uses a particular method since it is the patient’s persecuting voice which will gradually change in tone and become more and more admiring. Moreover, between sessions 4 and 5, the patients were instructed to question their loved ones to draw up a list of qualities which would then be questioned by the Avatar. Thus, it is possible that this process itself was therapeutic. Self-esteem and functioning have been shown to be linked in schizophrenia [46], and self-esteem has even been suggested as being a central component of recovery [47,48]. Moreover, a recent study showed that better self-reflectiveness might be linked to a better QoL [49]; therefore, the fact that this construct is thoroughly addressed during therapy could explain part of its efficacy.

Additionally, patients spoke more about their occupations following therapy (work/studies, hobbies, chores, etc.). Even though many organized activities were interrupted because of the pandemic, patients managed to start new hobbies and take care of themselves, as well as taking steps toward achieving their personal projects. For some patients, this even led them to make significant changes impacting their physical health (e.g., going to the gym, waking up earlier), or even to move in order to live in a home that better suited their needs. This finding, which was a priori not related to the initial goals of therapy, could also be linked to better self-esteem, thereby giving the patients the confidence to try something new or even to gain power over the symptoms, which would give the patients more freedom to get things done without being as bothered by them.

Some changes were also noted in the participants’ perceptions of their lifestyles. Notably, some of them reported having fewer addictive behaviors. While it should be noted that none of them suffered from a substance use disorder at baseline, the main issue for one participant was gambling. Pathological gambling is more common in patients with schizophrenia than in the general population, and it has recently been shown to be associated with psychotic symptoms, including hallucinations [50,51,52,53]. Although it was not a therapeutic target of therapy, ATp is flexible enough to allow participants to work on other goals if they wished to. Moreover, these components can be directly or indirectly linked to the voices; for example, gambling can trigger the voices, and the content of the voices can be related to drug consumption or gambling as well. For example, P2 explained that gambling would trigger his symptoms, which motivated him to stop. Therefore, it is possible that reduction in gambling behaviors translated to fewer symptoms, or conversely, that fewer symptoms resulted in fewer addictive problems; however, these hypotheses would need further investigation.

In schizophrenia, deficits in functioning can lead to social withdrawal and isolation, which perpetuates the vicious cycle of stigmatization that these individuals suffer from [54]. In a longitudinal study following 2010 schizophrenic patients over 6 months, it was even observed that social functioning was strongly correlated with psychotic symptomatology [55]. Therefore, social function could be an interesting indicator of treatment response. However, based on the content analysis performed in the present study, interpersonal relationships did not seem to change much following ATp. Nevertheless, one patient managed to make a new friend, to whom he became very close following therapy. Further investigation using validated tools will be necessary to assess the impact of therapy on these aspects.

Major life events can have a significant impact on QoL. However, the few participants who experienced such events between the two interviews stated that these events did not significantly impact their lives. Nevertheless, the COVID-19 pandemic is one specific event that impacted all participants. Not only are schizophrenia patients at increased risk for COVID-19 and related outcomes, but public health measures (e.g., physical distancing) could also worsen their psychosis prognosis [56,57]. Social distancing and canceled activities led many patients to social isolation, and fear of contracting the virus could also have an impact [56]. However, half of the participants mentioned that the pandemic had one or more positive aspects affecting their lives post-therapy. Given the difficult situation, it is even surprising to have been able to observe all the previously discussed impact of therapy. In a different context, it is possible that the observed impacts would have been slightly different, or even greater.

When asked about ATp specifically, all participants mentioned that they appreciated the intervention. Satisfaction with treatment has been shown to be associated with a better QoL in individuals with psychosis [18,58,59]; therefore, this might be an important component to consider. However, a few participants described some difficulties they encountered—notably, it could be difficult to role play a dialogue with a character. One participant also had trouble opening themselves to the therapist and talking about their feelings, which can indeed be intimidating. This could be related to the short duration of therapy, which could in certain cases not be sufficient to develop a strong therapeutic relationship. The differences between short-term and long-term psychotherapy have seldom been investigated; more research is needed on that matter [60]. Nevertheless, these difficulties only affected a minority of participants. Overall, most of them felt ATp helped them in some way, or that it helped reinforce some previously acquired knowledge. These changes could involve symptoms, new strategies to cope with symptoms, or they could be directly related to QoL (e.g., feeling better overall).

Although this qualitative study provides a deeper understanding of the evolution of QoL in patients who have undergone ATp, it has a few limitations that should be acknowledged. First, comparing the frequency of each theme before and after therapy did not allow consideration of the differences between both interviews. For example, the interviewer might have asked more questions about certain subjects during the baseline interview as they were getting to know the participant for the first time. This could explain why patients talked a bit less about their interests in the post-therapy interview, even though most participants mentioned having developed new interests or hobbies post-interview. Moreover, certain themes such as “Consumption/addictive behaviors” were not systematically explored, but often came up anyway during the interviews. Therefore, some participants might have displayed this type of behavior, but simply did not bring it up. Second, such a thorough, meticulous, and time-consuming analysis did not allow the analysis of a large number of participants, and thus the size of the sample was very small. Nevertheless, the study ensured that the profiles of these 10 participants were very similar to that of the entire sample who underwent ATp, by comparing their sociodemographic characteristics. Moreover, data saturation was reached; since adding more cases was not changing the results, analyzing more would probably not have changed them much either. Thirdly, certain obstacles to obtaining information were encountered in certain interviews, especially when the participants were very delusional or reluctant to confide. Nevertheless, these participants faced the same obstacles during both the pre-therapy and post-therapy interviews. Therefore, it is unlikely that the results regarding the frequency comparison were biased. In addition, most participants were lucid and open to sharing, which allowed the collection of rich information about their perspectives surrounding changes to their QoL following therapy. Fourth, there was no comparison group, which could have helped in assessing the change that might occur with the simple passage of time. However, these patients were all quite stable in their illnesses, since they had been diagnosed an average of 15.8 years before their baselines were determined, and they also had to be stable in their medication as well as in their mental and physical health to enroll in the clinical trial. Finally, this study took place during the onset of the COVID-19 pandemic, and the various lockdowns may have impacted the participants’ quality of life. In order to reduce this bias as much as possible, all changes directly related to the pandemic were classified in a separate theme.

## 5. Conclusions

In conclusion, this in-depth content analysis allowed a deeper understanding of the impact of ATp on the patients’ QoL. By analyzing the evolution of nine main themes and their many subdivisions three months after therapy compared to before therapy, it was possible to observe that ATp helped patients improve their psychotic symptomatology as well as their attitudes and perceptions of their symptoms, improve their self-esteem, develop new hobbies, find new projects, and take steps towards accomplishing them, as well as improve their overall lifestyles and moods. These results take us one step further into understanding how patients with long-standing treatment-resistant schizophrenia can still implement significant changes in their lives and progress towards recovery. These observations may well be considered when improving existing psychotherapeutic interventions or creating new ones, guiding future research, and improving our understanding of this population’s needs. For the future, it would be relevant to qualitatively analyze the impact of a more conventional psychotherapy (e.g., CBTp) on QoL and compare its effects to those of ATp.

## Figures and Tables

**Table 1 jpm-13-00522-t001:** Baseline characteristics. N = 10.

	Mean/N	%/SD
**Age**	38.2	12.5
**Gender**		
Male	7	70%
Female	3	30%
**Ethnicity**		
Caucasian	5	50%
Afro-American	3	30%
Other/Mixed	2	20%
**Civil status**		
Single	8	80%
In a relationship, without cohabitation	1	10%
Married	1	10%
**Employment status**		
Unemployed	7	70%
Employed	2	20%
Retired	1	10%
**Education**		
Completed high school	7	70%
Did not complete high school	3	30%
Years of education	12.5	3.3
**Principal Diagnosis**		
Schizophrenia	8	80%
Schizoaffective disorder	2	20%
**Psychiatric history**		
Number of hospitalizations	7.2	6.2
Age at illness onset	22.4	8.5
Time since illness onset (years)	15.8	7.9
**Antipsychotic medication**		
Clozapine	7	70%

SD: Standard Deviation; N/%: Number/percentage of subjects.

**Table 2 jpm-13-00522-t002:** Description changes following therapy for each participant.

Summary of Changes, Case by Case
**Case #1** was a woman in her late forties diagnosed with schizophrenia. She felt that AT helped her on many levels, especially regarding self-acceptance and her anxiety. She now felt better and had new projects; notably, she now wanted to take training to become a peer helper. Regarding her voices, her scariest one completely stopped, and she felt more equipped to deal with the remaining ones. Although she would still like them to go away, she now accepts the situation as it is.
**Case #2** was a man in his mid-forties diagnosed with schizophrenia. This patient affirmed that ATp provided him with many new tools that helped him face his hallucinations as well as having better self-acceptance and self-control. He also identified a link between his symptoms and his gambling problem as well as his anxiety. The voices are also much less frequent than before; he could now go a week or two without hearing any. Consequently, it was now much easier for him to hang out in public. His relationship with his family improved and he seemed no longer delusional.
**Case #3** was a woman in her late twenties diagnosed with schizophrenia. Although all her regular activities got canceled due to the pandemic, this patient still managed to start new hobbies and had new projects, which she was very excited about. She still heard her own thoughts just as before therapy and she was still unable to control them. Nevertheless, she had much less social anxiety than before and no longer had paranoid and guilt delusions. Moreover, she no longer had the vague suicidal thoughts that were present prior to therapy. Despite the restrictions, she still had a very good support network consisting of her family and her closest friends.
**Case #4** was a man in his mid-twenties diagnosed with schizophrenia. Following ATp, this patient made a new friend who was now taking an important place in his life. Indeed, he described himself as more social than before. He also had many new projects and took steps to accomplish them; for example, he started driving lessons to eventually drive his own car, which was a goal he had prior to therapy. Although the frequency greatly diminished, he still heard a voice that was trying to provoke him; however, he said that it no longer succeeded in making him angry like before. Moreover, he still had persecutory delusions.
**Case #5** was a man in his early thirties diagnosed with schizoaffective disorder. This patient affirmed that he felt much better than before ATp and had a new project involving a new friend of his. His relationships, hobbies and finances became stable. He reflected on his relationships and chose to stop maintaining some of them, mentioning that they were not bringing him anything positive. The frequency of his hallucinations greatly diminished and almost stopped during the past few weeks. He also got more control over them by successfully asking the voice to stop talking, whereas before they had control over him. He now felt “wiser, more apprentice, learned, positive, virtuous, moral”.
**Case #6** was a man in his early forties diagnosed with schizophrenia. This patient did not seem to evolve much since his pre-therapy interview, since he had the same objectives as before and did not take any steps to accomplish them. He was still hearing the voice of God, which he described as being very positive. From time to time, he also heard the devil’s voice whose content was negative. Like his pre-therapy interview, the post-therapy one was complicated and limited by many fixed thoughts and delusions, which made his QoL difficult to assess.
**Case #7** was a man in his early forties diagnosed with schizoaffective disorder. This patient found ATp difficult because of his difficulties in speaking about his symptoms, although he appreciated talking about them in the end. Nevertheless, he felt that the therapy did not change much for him. Indeed, his QoL did not seem much different than before therapy. He did start a new job, but only kept it for a few weeks, and therefore found himself unemployed again. However, he was motivated to find another job soon.
**Case #8** was a man in his early forties diagnosed with schizophrenia. This patient accomplished many objectives that he had prior to therapy, including moving to a bigger apartment. He explained that, after ATp, his voices became more intrusive and aggressive. Nevertheless, the voices eventually diminished a lot and became mainly positive as well. The therapy provided him with many new coping strategies that are still helping him make peace with his hallucinations.
**Case #9** was a man in his mid-twenties diagnosed with schizophrenia. This patient managed to achieve many of his pre-therapy goals, notably by moving into his own apartment and going back to school. He also had new professional and personal projects. Shortly after the end of therapy, he broke up with his girlfriend, which he described as being a very good thing; pre-therapy, he mentioned having relationship difficulties as she was being too dependent. Regarding his symptoms, he almost completely stopped hearing voices two months following therapy, a change that he attributed to his medication as well as to ATp. Consequently, he now had more time for himself and felt much better.
**Case #10** * was a transgender woman in her mid-sixties diagnosed with schizophrenia. Since she underwent therapy, this patient’s very aggressive hallucinations almost completely disappeared. Moreover, when that happened, she was now able to control them, which was not the case prior to therapy. Although she attributed these changes to ATp, it is important to note that her medication dosage increased during and after therapy. She still had many positive interpersonal relationships as she did prior to therapy and was participating in a new art-therapy group. However, she was now concerned about her recurring falls and the possibility of needing a cane to walk.

* Note: The results of this case need to be interpreted carefully as there was an unauthorized change in the medication dosage. For this reason, only the changes that the patient subjectively attributed to therapy were considered.

**Table 3 jpm-13-00522-t003:** Qualitative summary of changes by theme and subtheme.

Themes	Subthemes	Definition	Summary of Changes	Verbatims
Psychiatric symptoms	Psychotic symptoms	This category comprised hallucinations, mainly auditory verbal (e.g., voices), as well as delusions and other symptoms (e.g., thought disorganization, depression, anxiety). The intensity of symptoms, feelings aroused, coping mechanisms and their associated efficacy were also coded. Regarding the voices specifically, codes were created to characterize their content, omnipotence, and the nature of the relationship between the patient and their voice.	Regarding intensity, 8 participants reported that their symptoms improved after, which often made them feel better. Among them, 4 also had some form of recent improvement prior to therapy. The efficacy of coping strategies seemed to improve. Regarding the voices, overall, for many participants, the content was less negative and sometimes even became positive, the relationships between the patients and their voices improved, and beliefs about the voice being omniscient greatly diminished.	“After Avatar Therapy, I stopped hearing the voice.’’—P1, post-therapy “I feel better in terms of psychological health. I have less symptoms, less things psychologically aggressing me.”—P5, post-therapy “Since I underwent therapy, I put it… [The therapist], he helped me put him in his place.”—P10, post-therapy
	Impact of symptoms on QoL	Some patients described that the symptoms had an impact on their QoL, notably on their social behaviors, activities, and physical health.	In general, patients reported less impact of symptoms on QoL after therapy. Moreover, more participants reported that symptoms had little to no impact on their QoL following therapy.	“What I did not do before therapy that I can do now… I think it may be going to public places. […] Now, I can go places without difficulty.—P2, post-therapy
	Beliefs	All the beliefs the patients held towards their symptoms were coded as being insightful or lacking insight. This included beliefs regarding the origin of the symptoms, the existence of the voices, what schizophrenia is, if they identified with to their diagnosis, as well as gain of insight regarding previous delusions. Ambiguous utterances were excluded.	In general, the frequency of reporting beliefs was lower following therapy, regardless of insight. However, overall, fewer participants reported non-insightful beliefs post-therapy compared to baseline (4 vs. 8).	“Regarding schizophrenia, I believe I have more of a gift than a disease”—P1, pre-therapy “It all has to do with shame. The shame of being oneself, the shame of the negative thoughts.”—P1, post-therapy
Occupations	Work/studies	Verbatims related to work, school, volunteering, or entrepreneurship were coded as being either positive or negative. Neutral utterances were excluded. Stoppage or absence of occupation (e.g., being on social welfare) were rated as negative unless the situation was directly due to the COVID-19 pandemic.	In general, the working/studying situation did not change much, except for one participant who stopped working to go back to school (case 9).	“Ugh well my classes, I go to school… I’m lucky, I am in a program and then I can… go there part-time. It’s going well.”—P9, post-therapy
	Daily activities	Hobbies (i.e., individual or group activities, sport, relaxation) and chores (e.g., cooking, cleaning) were coded under this category. Discontinuation, abandonment, or absence of activities were also listed, but only when these were unrelated to the COVID-19 pandemic.	Although all participants reported having hobbies at baseline, the frequency of this theme increased, probably because many of them discovered new ones. Notably, P3 started cooking and adopted a dog that made her take more walks, P4 started going to the gym, P5 started playing guitar, P6 and P10 started art-therapy, and finally, P8 started photography. Although much less drastic, a similar increase in frequency was observed for chores.	“I almost never leave my house.”—P3, pre-therapy “I was doing Zumba, dance… And at [organization] we made sculptures in pottery like, clay, yeah. That was fun.”—P3, post-therapy
Interpersonal relationships	Improvement strategies	Some patients mentioned implementing strategies to improve their interpersonal relationships. For example, the patient might want to see close ones more often, or conversely want to distance themselves from certain friendships that bring more negative than positive.	Although this was not frequent, some participants were able to make meaningful changes in their relationships between both interviews.	“I reflected on my social circle of long-time friends. I just put two people aside. I am distancing myself. We have taken some distance over a few months, several months already.”—P5, post-therapy
Interpersonal relationships (continued)	Nature of relationships	Verbatims about the patients’ familial, amical or love relationships were classified as being either positive (i.e., regular contacts, offering support) or negative (i.e., conflicts or lack of relationship). Neutral utterances were excluded.	Overall, the number of verbatims referring to negative relationships slightly decreased.	“I left [ressource] because they said, I don’t know, there were complications there. They said I was scary, that I was scaring the neighborhood. I don’t know why they were saying that.”—P4, pre-therapy [About his new friend] “He is like a brother of blood. We smoke cigarettes outside. We talk.”—P4, post-therapy
	Perception of change	Some patients mentioned that their close ones have seen changes in them and their mental health. These changes were classified as being either improvements or deteriorations.	Although this theme was rare, following therapy, close ones saw improvements in 2 patients (cases 1 & 2) and deterioration was seen in case 6.	“They think I am improving. Less perfectionist, I take more care of myself, more structured, yeah.”—P1, post-therapy
Identity	Interests	Most patients expressed their interest for various subjects. This could be artistic, technological, related to sports, politics, business, finance, movies, a style of book or music, food, etc.	These did not vary much before vs. after therapy.	“I listen to Rock ’n’ Roll, dance music and all … dance mix, stuff like that. I like singing, stuff like that.”—P6, post-therapy
	Personality	All patients expressed their perception of their own personality traits. These were classified as being either positive (self-appraisal) or negative (self-deprecation). Neutral personality traits were excluded.	Notable changes were observed: 8 patients displayed way more self-appraisal post-therapy, and self-deprecation became less frequent.	“I accept myself, I love myself… I stop being a perfectionist and then I am more mature.”—P1, post-therapy
	Skills	Some patients affirmed that they possessed some skills, which could be for example social (e.g., ability to relate to others), basic life skills (e.g., manage their own finances) or self-management (e.g., be able to self-regulate emotions).	These were almost twice as frequent following therapy compared to before.	“I developed my communication skills”—P5, post-therapy
Identity (continued)	Relation to oneself	This category included feelings that patients maintained in relation to themselves. These could be conflictual (e.g., internal conflict) or, on the contrary, some patients mentioned achieving a form of self-reconciliation.	Notably, cases 1 and 2 mentioned that therapy helped on that matter.	“But hey, I accept my imperfections and then I try to continue to live well with myself and with others, without slapping myself on the wrist.”—P2, post-therapy
Wishes	Projects	This category included anything the patients wished to achieve. This could be a professional or educational objective, related to self-growth, weight loss, artwork, etc. Some participants also wished to remain stable, for example by keeping a job or an apartment for an extended period.	Although all patients mentioned having projects during both interviews, many new projects were present following therapy, resulting in a sharp increase in frequency.	“I want to continue [school] and maybe go to the gym with my sister. It would be beneficial for me.”—P9, post-therapy
	Steps taken	A distinction was made regarding whether the patients took steps toward accomplishing their objectives or not. These steps could be concrete actions, seeking information to do so, or even setting a date to carry out a project. Conversely, a verbatim was coded as “no steps taken” if the goal is vague or if the patient has no idea of what means they will use to get there.	Following therapy, more patients had taken steps toward accomplishing their project.	’But we have appointments and all, so I don’t know, it’s like…”—P3, pre-therapy. “I am learning how to drive at the moment … it’s going well, my exam is January 12.”—P3, post-therapy
Lifestyle	Consumption/addictions	Although this was not systematically questioned since a diagnosis of substance use disorder was an exclusion criterion, several patients mentioned consuming alcohol, cannabis, or illicit drugs. Pathological gambling was also present for P2. Some positive or negative impacts associated with substance use or addictive behaviors were identified, as well as some coping strategies or reasons to reduce, stop, or avoid relapse in previously addicted patients.	Overall, all addiction-related verbatims were less frequent post-therapy compared to baseline.	“It’s a problem that I have with gambling. This anxiety … when it’s there, I kind of have suicidal gestures, in fact. I tell myself, well, that’s how it is, I’m going to put my money in gambling, I don’t care.”—P2, pre-therapy
Lifestyle (continued)	Physical disorders/symptoms	Patients with schizophrenia often have many physical comorbidities or symptoms, which were generally discussed during the interviews. Each verbatim referring to physical health was classified as being negative (pain, dissatisfaction with sleep or sexual health, weight gain/loss, illnesses, medication side effects, physical symptoms, and their impact) or positive (i.e., having a good sleep schedule or sexual life). Neutral utterances were excluded.	The frequency and number of patients mentioning health issues did not change much through the follow-up. However, more patients seemed to have a better perception of their health, with four patients saying good things about their physical health post therapy (6 verbatims) as opposed to one prior to therapy (1 verbatim).	“I worked on this with [the therapist] and I am no longer ashamed. I like my libido.”—P1, post-therapy
	Mood	This category included the patients’ general mood, state of mind, or any verbatim reflecting how the patient is currently feeling. These utterances were classified as positive/good, or negative/bad.	It was possible to observe that, following therapy, more patients displayed a good mood and less mentioned having a bad mood.	“For sure, I’m calmer now than before therapy”—P2, post-therapy
	Housing	Patients often expressed how they felt about housing or lack of housing. This could generally be classified as positive (satisfied) or negative (unsatisfied); neutral utterances about housing were excluded.	Following therapy, dissatisfaction with housing were less frequent because of cases 8 & 9, both of whom moved into new apartments that better fitted their needs.	“I am moving into an apartment soon. I stayed here for around 6 years. I am still improving, but it helped me a lot to come here.”—P9, pre-therapy “Since I’m in an apartment, I feel a lot… I feel free and it makes me feel good.”—P9, post-therapy
Psychiatric care	Avatar Therapy	At the very end of the second interview, participants were questioned about their ATp experience.	All of them stated that the therapy helped in some way. Most of them expressed some form of appreciation, and a few mentioned that therapy helped reinforce previously acquired knowledge. In some cases, difficulties were also encountered.	“I’m happy I did it because, like I said, it helped me control my voices, and I found it interesting. The psychologist, he helped me understand stuff regarding my voices.”—P9, post-therapy”I am able to accept myself and not stop doing things just because the voices are against it. It was almost always present before, but this is something I have much improved with therapy.”—P2, post-therapy
	Follow-up by the treatment team	When talking about their psychiatric follow-up outside ATp, patients could have positive or negative experiences. As the patients’ perception of their care can change, these were documented under this category. Neutral utterances were excluded.	In general, negative comments regarding psychiatric treatment or treating teams were less frequent following therapy, while positive ones remained the same.	“I still have those hallucinations. What do I do? They tell me … they kind of remained vague, a bit. They just told me it was my illness. I say that I took the medication for so many years… It can’t be just that.”—P5, pre-therapy
Life events	Major negative events	Any current or recent major negative event was documented under this category.	These events, present for 2 participants before therapy and 3 participants after therapy, needed to be considered because these could impact changes that were observed on QoL. First, in the case of P1, her father was sick pre-therapy, and subsequently died 2 weeks before the post-therapy evaluation. However, she was mostly relieved by this since his suffering finally came to an end. Moreover, prior to therapy as well as after therapy, P6 mentioned that he was recently confronted by drug dealers who meant to harm him. The temporality as well as the possible delusional nature of these altercations remained unclear. Finally, P10 expressed some concerns regarding her sister’s mental health during the post-therapy interview. All in all, these events did not seem to have much impact on the participants’ interviews.	“My dad has Alzheimer, he is 84 years old and he is bedridden in a long-term care facility.”—P1, pre-therapy “It’s ok. I’m happy that he’s gone because he has been bedridden for 10 years and it was horrible. So I feel very relieved.”—P1, post-therapy
	COVID-19 pandemic	As this study was mostly carried out during the COVID-19 pandemic, all effects that the patients attributed to the pandemic were documented under this category. This was done with the objective of ensuring that the variations captured in the other themes are not directly attributable to the global health situation. A distinction was made between the positive aspects of the pandemic (or good adaptation) and the negative ones.	Overall, COVID-related themes were more frequent post-therapy, suggesting that the pandemic could have had an impact on observed results. However, good adaptations and positive aspects were only observed post-therapy.	“I have been teleworking since April. It saves me an hour a day of traveling, so I’m not complaining. I enjoy these moments.”—P2, post-therapy”The voice hearers’ group has been canceled. I had many activities, and they were all canceled.”—P1, post-therapy
Attitudes/behaviors during the interview	Presence of hallucinations	From time to time, the presence of hallucinations during the interview could hinder the course of the discussion.	These were much more present during the pre-therapy interview, but this is almost exclusively explained by case 10 who had a lot of them.	“I hear it, I hear it coming. Shhhh. Leave me alone. You’re annoying, you’re annoying, you’re annoying.”—P10, pre-therapy
	Patient cooperation	While some patients were uncooperative/closed (e.g., stated that they did not want to talk about something), others were very cooperative and had an open mindset, for example by bringing notes, by using humor, or by asking questions.	Overall, cooperation did not vary much over the two interviews.	“If there is anything else I can do, I am available to do it.”—P9, post-therapy
	Confusion	The course of the interview could sometimes be hindered by the patient becoming confused, having memory loss, misunderstanding the evaluator’s question, lacking attention, or only talking about a few specific subjects.	Although the level of confusion overall decreased, it drastically increased for case 6, which greatly hindered the interview process.	“I was beautiful when I was younger. But there, I don’t know what I did. I was like, oh, that’s not good, all the time. I was seeing the people, they were losing their teeth because they consumed drugs, stuff like that”—P6, post-therapy

**Table 4 jpm-13-00522-t004:** Qualitative changes in quality of life following Avatar Therapy for psychosis. N = 10.

Codes	Freq. Baseline	Freq. Post ATp	N Baseline	N Post ATp	Variation Score
Psychiatric symptomatology					
Improving symptoms	16	59	6	8	6
Worsening symptoms	4	5	3	4	1
Positive feelings	0	6	0	5	5
Negative feelings	57	27	10	5	−7
Improving feelings	1	7	1	5	5
Active strategies	63	53	10	10	1
Passive strategies	13	23	6	7	3
No strategies	4	0	3	0	−3
Effective coping mechanisms	8	9	3	5	0
Sometimes/moderately effective	12	5	7	3	−3
Ineffective coping mechanisms	8	3	5	2	−4
Positive voice content	6	13	4	4	2
Negative voice content	33	14	9	5	−5
Positive relationship w/the voice	6	11	3	5	2
Negative relationship w/the voice	41	21	5	5	−3
The voice is powerless	9	15	4	5	3
The voice is omnipotent	13	6	5	3	−3
Thought disorganization	5	3	2	1	−1
Anxious symptoms	5	9	2	2	0
Depressive symptoms	5	3	3	2	−2
Grandiosity delusion	7	2	3	2	−3
Persecutory delusion	28	3	6	2	−4
Other delusion	45	2	8	1	−8
Impact on social behaviors	16	4	6	3	−4
Impact on activities	7	2	3	1	−2
Impact on physical health	9	4	4	4	−2
Little to no impact on QoL	3	5	2	5	2
Beliefs: insight	38	21	8	8	−4
Beliefs: lack of insight	14	10	8	4	−4
Occupations					
Positive work/studies	6	14	3	3	1
Negative work/studies	1	6	1	5	4
Hobbies	39	74	10	9	6
Chores	12	21	7	7	2
Discontinuation of activities	6	7	4	3	0
Interpersonal relationships					
Improvement strategies	7	8	3	2	−2
Positive relationships	79	88	10	10	0
Negative/lack of relationships	64	39	9	9	−2
Close ones see improvement in patient	0	6	0	2	2
Close ones see deterioration in patient	0	1	0	1	1
Identity					
Interests	29	25	7	6	−3
Self-appraisal	8	22	5	8	6
Self-deprecation	29	11	6	4	−2
Skills	16	34	7	8	3
Reconciliation with oneself	3	15	2	3	1
Conflictual relationship with oneself	4	3	2	1	0
Wishes					
Projects	46	76	10	10	6
Steps taken	10	29	3	8	4
No steps taken	0	5	0	3	3
Lifestyle					
Consumption or addictive behaviors	19	3	6	2	−5
No consumption/addiction	7	5	6	5	−2
Strategies related to consumption	8	5	3	1	−1
Positive impact of consumption	5	2	2	2	−2
Negative impact of consumption	14	5	4	2	−3
Health issues	48	40	7	7	1
Good health	1	6	1	4	3
Good mood	15	24	7	8	5
Bad mood	25	11	6	4	−3
Satisfied w/housing situation	3	4	2	1	0
Unsatisfied w/housing situation	10	1	2	1	−2
Psychiatric care					
Appreciation of ATp	n/a	30	n/a	10	n/a
Feeling that ATp helped	n/a	36	n/a	9	n/a
It reinforced previously acquired skills	n/a	2	n/a	2	n/a
Patient encountered difficulties with ATp	n/a	13	n/a	4	n/a
Positive experiences with treatment	14	13	6	5	0
Negative experiences with treatment	15	5	5	2	−4
Life events					
Major negative event	3	8	2	3	1
Good adaptation to COVID/positive	0	8	0	5	5
Bad adaptation to COVID/negative	10	22	5	6	3
Attitudes and behaviors during the interview					
Presence of hallucinations/delusions	26	3	2	2	0
Open attitude/cooperative	10	6	5	5	−1
Closed attitude/uncooperative	10	5	3	3	−1
Confusion	38	63	8	6	−2

## Data Availability

Not applicable.

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
