# Peer review of "Changes in Quality of Life in Treatment-Resistant Schizophrenia Patients Undergoing Avatar Therapy: A Content Analysis"

_jpm, 2023, doi:10.3390/jpm13030522_

Round 1
Reviewer 1 Report
I would like to say thanks for the opportunity to review this article.
The article presented has a very interesting theme and important to the scientific community and community in general.
It presents the impact on quality of life in patients with schizophrenia undergoing Avatar Therapy. I must congratulate the authors by the implementation of these therapy.
Overall, the article has a scientific and appropriate writing, including all the components of a good scientific research. The title and abstract are related with the content. The keywords are linked to the research but most of them are not included in Mesh Terms.
The article has chapters organized in a logic way. Introduction allows the framing of the theme and the research itself. The main goal is appropriate. Methodology is scientifically appropriate, and completely exposed. Results are adequate and complete. Some tables area a little to extensive, which can disturb the reading and data analysis. Discussion is done according to the results and allows to understand the response to the main goal. Authors present limitations of the study and has appropriate conclusions. References are recent in their majority (about 50% have less than 5 years, and around 25% have more than 10 years) and are pertinent.
For this, we suggest the following corrections:
- We suggest finding more appropriate and indexed keywords
- The selection process of the patients to interview it is not clear. How many patients participated in the trial, and why were considered 10 to interview? Why were those selected and not others?
- the first paragraph of results must be removed or replaced by a relevant information (lines 214, 215 and 216)
- it should be included an explanation about ethical issues regarding the procedures of the interviews.
- there is no clarification about why the medication analysis included clozapine as a category itself and the other antipsychotics were grouped in typical and atypical.
Congratulations.
Thank you.
Author Response
The authors would like to thank the reviewer for their time as well as their kind words regarding the manuscript. It is always a pleasure to read such a positive review from experts in the field. Please find below our answers to the following corrections that were suggested. 1. We suggest finding more appropriate and indexed keywords Authors’ response: Thank you for this helpful suggestion. The AVH and TRS acronyms have both been removed. 2. The selection process of the patients to interview it is not clear. How many patients participated in the trial, and why were considered 10 to interview? Why were those selected and not others? Authors’ response: We would like to thank the reviewer for this observation, as it is true that this aspect of the methodology was not clear in the manuscript. The included subjects were simply the first 10 participants to have completed ATp as well as the post-therapy interview, as these were the only ones available at the time the analyses were conducted. In fact, the cases were transcribed and analyzed as soon as the second interview was completed. This information has been added to the “Material and Methods” section, lines 124–126. 3. the first paragraph of results must be removed or replaced by a relevant information (lines 214, 215 and 216) Authors’ response: We would like to thank the reviewer for this mindful observation, as it was indeed a mistake. This paragraph has been removed. 4. it should be included an explanation about ethical issues regarding the procedures of the interviews. Authors’ response: Thank you for this suggestion. It is true that these in-depth interviews may arouse unpleasant memories or feelings in some participants, and measures have been put in place to support them as much as possible (e.g., phone support by a qualified nurse). This information has been added to the “Material and Methods” section, lines 163–166. 5. there is no clarification about why the medication analysis included clozapine as a category itself and the other antipsychotics were grouped in typical and atypical. Authors’ response: Thank you for this comment. Lack of adequate treatment response despite the use of clozapine is often referred to as “ultra-resistant schizophrenia”. In recent literature, it has been suggested that these patients may have specific characteristics distinct from those of patients with regular treatment-resistant schizophrenia, hence why we decided to present this result. This information has been added to the manuscript, see line 226. As for the atypical vs. typical nomenclature, it is commonly used as these two types of antipsychotics have different pharmacodynamic properties. It is true that, for this study, this distinction does not add much information regarding the characteristics of the sample. Therefore, these two lines have been removed from Table 1.

Reviewer 2 Report
The work is extremely interesting. To improve the perception of the text, I would recommend a more detailed description of the content analysis procedure. Were all patients included in the studies? Were there patients who declined to participate? This may be important.
The use of acronyms needs to be reconsidered. Sentences such as "ATp is a novel psychotherapy using VR to treat AVH in TRS" are perceived only by specialists.
Author Response
We would like to thank Reviewer 2 for their time and useful suggestions. Please find our answers to your comments below.
- The work is extremely interesting. To improve the perception of the text, I would recommend a more detailed description of the content analysis procedure. Were all patients included in the studies? Were there patients who declined to participate? This may be important.
Authors’ response: Thank you for this useful observation, as it is true that this aspect of the methodology was not clear in the manuscript. The included subjects were simply the first 10 participants to have completed ATp as well as the post-therapy interview, as these were the only ones available at the time the analyses were conducted. In fact, the cases were transcribed and analyzed as soon as the second interview was completed. None of the participants included in the main clinical trial refused to participate in this particular subproject. This information has been added to the “Material and Methods” section, lines 125–126 and 133–135.
- The use of acronyms needs to be reconsidered. Sentences such as “ATp is a novel psychotherapy using VR to treat AVH in TRS” are perceived only by specialists.
Authors’ response: We would like to thank the reviewer for this helpful comment. The AVH and TRS acronyms have been removed.

Reviewer 3 Report
Dear authors, thank you for giving me the opportunity to review your manuscript: “Changes in quality of life in treatment-resistant schizophrenia patients undergoing Avatar Therapy: A content analysis”.
I send below some comments to the manuscript.
Introduction:
- The introduction is well-written, well-structured, and quite clear.
Materials and Methods
- This section is well-written and clear.
Results
- The results are clear and well structured.
Discussion:
- The authors discuss the results and indicate the limitations of the study.
The topic covered is relevant and can bring important information to the therapeutic change process.
The authors present a good research paper, which may have several implications for practice.
Author Response
The authors would like to thank the reviewer for their time and kind words.
